# Alternative Additives for Organic and Natural Ready-to-Eat Meats to Control Spoilage and Maintain Shelf Life: Current Perspectives in the United States

**DOI:** 10.3390/foods13030464

**Published:** 2024-02-01

**Authors:** Aaron R. Bodie, Lindsey A. Wythe, Dana K. Dittoe, Michael J. Rothrock, Corliss A. O’Bryan, Steven C. Ricke

**Affiliations:** 1Department of Animal and Dairy Sciences, University of Wisconsin, Madison, WI 53705, USA; aaronb@werfoodsafety.com (A.R.B.); lawythe@tamu.edu (L.A.W.); 2Department of Animal Science, University of Wyoming, Laramie, WY 82071, USA; ddittoe@uwyo.edu; 3Egg Safety and Quality Research Unit, U.S. National Poultry Research Center, United States Department of Agriculture-Agriculture Research Service (USDA-ARS), Athens, GA 30605, USA; michael.rothrock@usda.gov; 4Department of Food Science, University of Arkansas-Fayetteville, Fayetteville, AR 72701, USA; cobryan@uark.edu

**Keywords:** RTE foods, alternative additives, clean-label, spoilage, antimicrobials

## Abstract

Food additives are employed in the food industry to enhance the color, smell, and taste of foods, increase nutritional value, boost processing efficiency, and extend shelf life. Consumers are beginning to prioritize food ingredients that they perceive as supporting a healthy lifestyle, emphasizing ingredients they deem acceptable as alternative or “clean-label” ingredients. Ready-to-eat (RTE) meat products can be contaminated with pathogens and spoilage microorganisms after the cooking step, contributing to food spoilage losses and increasing the risk to consumers for foodborne illnesses. More recently, consumers have advocated for no artificial additives or preservatives, which has led to a search for antimicrobials that meet these demands but do not lessen the safety or quality of RTE meats. Lactates and diacetates are used almost universally to extend the shelf life of RTE meats by reducing spoilage organisms and preventing the outgrowth of the foodborne pathogen *Listeria monocytogenes*. These antimicrobials applied to RTE meats tend to be broad-spectrum in their activities, thus affecting overall microbial ecology. It is to the food processing industry’s advantage to target spoilage organisms and pathogens specifically.

## 1. Introduction

Generally, food additives are used in the commercial food industry to improve foods’ color, smell, and taste, enhance nutritional value, increase processing efficiency, and extend shelf life [1,2]. Food additives are a broad group of compounds that potentially improve food quality [3]. Traditional food additives include salt, sugar, herbs or spices, vinegar, sulfur dioxide, and colorants [3]. However, the demand for types of food additives has changed with more emphasis on alternative additives deemed acceptable as “clean-label” [4]. The clean-label market trend for foods has become increasingly important to consumers in Western countries, and the onset of the pandemic in 2020 provided further motivation for consumers to prioritize food ingredients that they perceive as supporting a healthy lifestyle [5]. There is no legal or regulatory definition of a “clean-label” [4]. Consumers may equate clean-labels with minimally processed foods with no artificial flavors, artificial colors, or synthetic additives [4].

Consequently, food ingredients viewed as “healthier” or providing health-like benefits have gained commercial interest in the retail marketplace. These ingredients include those identified as part of a “clean-label” type of classification. A broad definition of “clean-label” foods is those with no artificial additives and are less processed, with simple and short ingredient lists that the average consumer recognizes [6,7]. Reduced fat, sugar, and sodium contents may also fall under the term “clean-label” when contributing to improved health [7]. 

In general, consumers rely on informational statements on packages such as “no preservatives” or “reduced fat” rather than reading the ingredient statements; for example, it has been reported that 65% of respondents to surveys responded that they never checked ingredient labels for additives [8], and some estimate that only 5 to 10% of consumers read the ingredients [9]. Delgado-Pando et al. [10] believe that consumers associate clean-label products with those with a short list of nonsynthetic or familiar ingredients. The Institute of Food Technologists (IFT) has proposed a definition: “clean-label means making a product using as few ingredients as possible and making sure those ingredients are items that consumers recognize and think of as wholesome” [11]. Cao and Miao [12] surveyed consumer-perceived clean-label attributes and found that the top three were minimally processed, eliminated undesired ingredients and exercised humane treatment. In order of perceived importance, they were minimally processed, had familiar ingredients, eliminated undesired ingredients, used local ingredients, and demonstrated an absence of allergens [12]. 

Clean-labeling in meat products, particularly further processed meats such as ready-to-eat (RTE) meats, has also become an issue. Most antimicrobials that are applied to RTE meats tend to be broad-spectrum in their activities, so it would be expected that overall microbial ecology would be impacted by their administration [13]. This, in turn, could alter the extent of shelf life and subsequent development of quality deterioration identified as spoilage. Historically, this has been difficult to characterize with conventional microbial culture-based approaches due to the limitations and selectivity of most media used for cultivation and enumeration. While some microorganisms have been identified as potential perpetrators and indicators of spoilage, these associations are generally relatively imprecise. Current and potential additive candidates for clean-label RTE applications in the United States (U.S.) are discussed in this review, along with their interactions with the microorganisms present in these RTE meat products. 

## 2. Labeling Basics

Ready-to-eat meat products risk microbial contamination after thermal processing, leading to food quality loss and increased risk to consumers’ health [14]. The desire for food with no artificial additives or preservatives has led to research on additives that are acceptable to consumers but do not lessen the safety or quality of RTE meats. The United States Department of Agriculture (USDA) has specific regulations requiring meat processors to employ measures to control *Listeria monocytogenes* when products are exposed to the environment after cooking, such as slicing a ham after it is cooked [15]. Lactates and diacetates have been used almost universally to extend the shelf life of RTE meats by reducing spoilage organisms and preventing the outgrowth of *L. monocytogenes* [15]. Despite the increased safety these chemicals provide, they are not acceptable to consumers who desire a clean-label, and several retailers and food service operators have declared they do not want products containing lactate or diacetate [16].

### 2.1. Natural as a Food Label in the U.S.

Currently, the U.S. Food and Drug Administration (FDA) does not have a formal definition for “natural” as a food labeling term, although there is a policy that states “natural” on a label means that nothing artificial or synthetic has been added [17]. This policy does not address food production, processing or manufacturing methods, or the perception of a food’s nutritional or health benefits [17]. The U.S. Department of Agriculture (USDA) considers a natural meat and poultry product as “containing no artificial ingredient or added color and is only minimally processed” [17]. When using the term “natural” on a label, processors must also include a brief explanation, such as “no artificial ingredients” or “minimally processed”. Consumers tend to view products labeled “all-natural” as higher quality and more nutritious [18]. Chambers et al. [19] questioned a group of consumers on which compounds on a list of ingredients they would consider “natural”; around half of the respondents considered corn, wheat flour, black beans, soybeans, sugar, and salt as natural. Highlighting inconsistencies in consumer perception, wheat flour was identified as “natural” by nearly three times as many people as gluten, which is the major protein in wheat flour [19]. When considering baking soda versus sodium bicarbonate (the chemical name of baking soda), baking soda was more likely to be considered natural [19]. When asked what they thought the term “natural” meant, they answered that they believed it to mean the absence of additives or minimal human intervention [20]. Roman et al. [21] conducted a systematic review of research and concluded that “naturalness” is high-ranking worldwide, regardless of when the research was performed. However, researchers defined “naturalness” differently, so comparing the qualities of a food or ingredient that consumers believe is natural was problematic. 

### 2.2. Organic Labeling in the U.S.

Organic farming is defined by the USDA as agricultural production that “promotes and enhances biodiversity, biological cycles, and soil biological activity” by using “materials and practices that enhance the ecological balance of natural systems” [22]. Processing techniques, including curing, synthetic preservatives, and synthetic antimicrobials, do not automatically disqualify a product from being labeled as organic so long as the additive follows the guidelines laid out by the USDA (Table 1). Antimicrobials and additives allowed for organic products do not necessarily align with accepted-label antimicrobials. 

### 2.3. Alternative Additives in RTE Meat Products: The Nitrate Story

Adding nitrates and other curing agents in RTE meats has historically been a foundational aspect of reducing pathogens, such as *Clostridium botulinum*, reducing lipid oxidation, and promoting the color fixation associated with products [29,30]. Nitrates and nitrites are commonly used in packed RTE meat products and have proven effective in inhibiting pathogens such as *Clostridium perfringens* [31]. *Cl. perfringens*, a Gram-positive spore former, can form highly thermo-resistant spores that can become vegetative during temperature abuse following thermal processing. However, consumers who seek out clean-label foods no longer believe that the addition of nitrates is acceptable because, for years, nitrates/nitrites were thought to be a risk for causing cancer since they can be converted into nitrosamines that have carcinogenic effects [32]. 

Most striking about this consumer-led rejection of nitrite/nitrate additives is the overwhelming presence of nitrates and nitrites naturally occurring in fruits and vegetables, up to 80% of dietary sources (Table 2) [32]. These include fruits and vegetables in heavily promoted dietary regimens perceived as healthy, such as the Dietary Approaches to Stop Hypertension (DASH) diet and the Mediterranean diet [33]. Indeed, evidence suggests that there may be physiological importance and even health benefits to nitrites and nitrates in the diet, especially regarding their reduction to nitric oxide (NO) [32,33,34]. Evidence shows that dietary nitrates and nitrites decrease blood pressure by converting nitrates to nitrites and then to NO [35,36,37,38,39]. Proposed effects include regulating cardiovascular health, blood pressure, and hypoxic vasodilation during exercise; increasing mucosal blood flow and mucosal thickness; and implications for Alzheimer’s Disease and other aging-related diseases [40,41,42,43,44]. With these potentially significant implications, adding nonsynthetic, plant-based extracts that contain nitrates and nitrites appears as a logical next step for achieving the clean-label goal in pathogen and quality control in RTE meats. However, some clean-label-oriented consumers reject the presence of even these nonsynthetic sources of nitrates and nitrites [45]. In 2019, Consumer Reports conducted a telephone survey of 1000 U.S. adults; they reported that 46% of people who consumed deli meats looked for ones labeled as “no nitrates added” [46]. Respondents were also asked about the “no nitrates added” label, with an explanatory statement that only naturally occurring nitrates were used, and 42% said they would be confused about whether nitrates were added or not [46]. 

## 3. Bacterial Ecology and Spoilage of RTE Meats 

The USDA Food Safety and Inspection Service (FSIS) defines RTE meat and poultry products as being fully cooked before packaging and not requiring further cooking by the consumer [47]. Some examples of RTE products are hot dogs, luncheon meats, cold cuts, fermented or dry sausage, and other deli-style meat and poultry.

### 3.1. Raw-Cured Shelf-Stable Meats

This group includes raw hams and dry sausages such as Soppressata, Salami, air-dried Pepperoni, Cerevelat, and Genoa. *Micrococcaceae* are the most abundant microbiota detected on hams during curing and drying; 60% of the isolates belong to the genus *Staphylococcus*, with 25% being *Micrococcus*, with the rest divided among *Kocuria, Dermacoccus*, and *Stomatococcus* [48]. The major species identified are *Staphylococcus xylosus*, *Micrococcus lylae*, *Staphylococcus equorum*, and *Dermacoccus nishinomiyaensis* [48,49,50,51]. Small numbers of enterotoxin A producing *S. aureus* can be identified on ham during its production, but it is not detected on correctly formulated dried raw ham [52]. *Penicillium* spp. molds were found on hams in the first stages of production, but these were later replaced by *Aspergillus* spp. [53,54].

Hams are cured by applying dry salt, pumping brine into the meat, or soaking in brine [55]. This initial cure is performed at a temperature of less than 5 °C. Spoilage of raw hams can be caused by species of *Enterobacteriaceae* and *Clostridium* that can grow in the hams before the salt concentration becomes high enough or if the curing temperature is too high to prevent multiplication [56]. *Cl. botulinum* can produce toxins at temperatures as low as 8 °C [57]. To prevent access to these organisms, the carcass from which the hams are derived must be appropriately chilled using clean equipment, brine, and optimal temperature control [56]. If stored under high humidity, these meats can develop surface mold and become rancid depending on storage time/temperature [55]. 

### 3.2. Dried Meats

Dried meats have a very low water activity, preventing the growth of bacteria at room temperature [58]. Dried meats are used in soup mixes and snack foods [55]. A typical dried meat snack food is beef jerky, the commercial production of which includes a cooking step [59]. There are three ways of making beef jerky. In the first procedure, beef is trimmed, injected with a cure, massaged, put in casings, and cooked to an internal temperature of approximately 65 °C. After cooling, logs are sliced into disks, which are dried and smoked [60]. The cooking step kills the vegetative bacteria on the raw meat, but recontamination can occur during handling, slicing, and loading into the drier.

In the second procedure, raw beef is sliced into strips, soaked in the cure, and then dried and smoked. In the third method, the meat is flaked with the cure ingredients and then formed into logs, which are subsequently chilled. These chilled logs are sliced, dried, and smoked. In these last two processes, bacteria from the raw meat can still be in the slices [60]. For all three methods, it must be ensured that the slices are fully separated for the drying process and that the drying takes place rapidly to keep the inherent bacteria from growing. The water activity is reduced to approximately 0.86 in 1 to 3 h and to a final water content of 15 to 20%. The USDA has established a moisture/protein ratio requirement of 0.75 for beef jerky [59]. Mold and yeasts are the primary spoilage organisms in dried meats, especially if they can absorb moisture from the air; this is prevented by drying to the required low water activity and vacuum packing or by drying and keeping the water activity below 0.70 [60].

Asefa et al. [61] collected samples from dried meat products and the environment of a processing plant to assess levels of yeasts and molds. They obtained 901 isolates, 500 (57%) of which were molds, and 401 (43%) were yeasts [54] (Asefa et al., 2010). The yeast isolates were identified as *Debaryomyces, Candida*, *Rhodotorula*, *Rhodosporidium*, *Cryptococcus*, and *Sporidiobolus* [62]. Over 39 mold species were detected, with almost 80% of these belonging to the species of *Penicillium* [61].

### 3.3. Fully Retorted Shelf-Stable Uncured Meats

These products include soups and other products stored in cans or pouches or aseptically packed into sterile containers after cooking [63]. The vegetative cells usually present on raw meats are easily killed by heat processing, and the product will not usually spoil if stored at less than 40 °C [64]. Retorted foods can be contaminated after processing if pin holes develop in cans or pouches that have not been sealed correctly or by rough handling during shipping and storage; this usually results in a mixed culture of spoilage organisms [55].

Proper heat treatment kills all vegetative bacteria, but if the product were spoiled before thermal processing, the remaining metabolites would still be unacceptable. Spores of putrefactive bacteria such as *Cl. sporogenes* are highly heat-resistant. If thermal processing is not adequate, surviving spores may germinate and cause spoilage during storage at ambient temperatures [55]. If cans or pouches have pin holes or compromised seams, bacteria can enter the container after processing; this type of spoilage is usually caused by mixed cultures [55]. 

### 3.4. Cooked Perishable Cured Meats

This category includes cold cuts, deli/luncheon meats, charcuterie, pate, frankfurters, and bacon [65]. Cooked perishable cured meats contain approximately 125 mg/kg of nitrite, are cooked to within a range of 65 to 75 °C (149 to 167 °F), require refrigerated storage, and may or may not be heated before consumption [55]. In North America, bacon is cooked before serving. Some cured meats (corned beef, ham, and bacon) are made from intact muscles and cuts of meat or from pieces of meat that have been massaged and tumbled and then formed in casings or molds [55]. Others, such as sausages, pate, and frankfurters, are made from comminuted meats extruded into casings or molds [55]. A cure, containing sodium nitrite, is added to the meat by injection of brine, soaking in brine, or blending during emulsion preparation [60]. Any bacteria present on the surface of meats will be uniformly dispersed throughout the product during cure injection, reforming, or emulsion preparation [49]. The processing of these products normally kills parasites and vegetative cells. However, some heat-resistant *Streptococci* will survive [55], as well as some strains of lactic acid bacteria, such as *Lactobacillus viridescens* [66]. The spores of *Clostridia* and *Bacillus* will not be affected by the level of heat used for these products [55].

These perishable products must be kept refrigerated, as usually stated on the package. Canned hams sold in their cooking containers are considered semi-perishable and will not spoil unless held above 10 °C (50 °F) [55]. However, if the products are repackaged or sliced, they are exposed to environmental contamination and spoilage [55]. This spoilage is indicated by off odors, off flavors, discoloration, or slime formation; a bacterial biomass above 7 log CFU/g is generally considered sufficient to cause spoilage [67]. Typically, spoilage of meat and poultry products is caused by psychrophilic *Pseudomonadaceae*, *Enterobacteriaceae*, *Listeriaceae*, and *Lactobacilliaceae* [68]. 

The spoilage rate at refrigeration temperatures is influenced not only by the product’s composition but also by the packaging and storage temperature [69,70]. Spoilage during refrigeration can be caused by enterococci that survive heat processing [71] or by *Clostridium putrefaciens*, particularly in products with low salt content [69]. 

Souring, discoloration, milky exudate, and slime characterize the spoilage of products repacked in low gas permeability [72]. These defects are caused by lactic acid bacteria, including *Lactobacillus* or *Leuconostoc* [72]. *Leuconostoc* produces a dextran slime [73]. Lactic acid bacteria produce hydrogen peroxide that oxidizes the porphyrin ring of nitrosohaemochrome to choleomyoglobin, thus causing greening [74]. Any products removed from casings and repackaged or sliced and repackaged are subject to post-process contamination. Contaminants can include psychrotrophic lactic acid bacteria [66], *Brocothrix thermosphacta*, *Listeria* spp., and *Enterobacteriaceae* [75]. 

However, as Huang et al. [76] concluded, understanding the metabolic pathways that lead to spoilage in meat is necessary to control spoilage. These spoilage microorganisms utilize different mechanisms according to the type of food, and interaction with other microorganisms will accelerate the spoilage process. To better understand the spoilage functions of these microorganisms in foods, using high-throughput techniques such as de novo sequencing is valuable. Recent developments such as predictive biology [77] and metagenomics, meta-transcriptomics, and metabolomics [78] offer opportunities to expand this knowledge. 

## 4. Clean-Label Antimicrobials in the U.S.

### 4.1. Organic Acids

Organic acids and salts of acids are theorized to exhibit antimicrobial activity in their un-dissociated and uncharged forms by being able to cross the bacterial cell membrane [79], where they increase osmotic stress and disrupt biomolecule synthesis, leading to bacterial death [80,81,82]. Organic acids have shown promise as natural antimicrobials but may affect color and flavor changes in products [83]. The most used acids are acetic, citric, lactic, propionic, malic, succinic, and tartaric [84,85]. Their many advantages include GRAS (generally regarded as safe) status by the U.S. Food and Drug Administration (FDA), no limits on acceptable daily intake, low cost, and ease of use, but caution must be observed as some *Salmonella* strains can develop resistance to acidic conditions [86]. 

Sodium lactate, the salt of lactic acid, has been used as an antimicrobial in Chinese-style sausages; lower total aerobic and anaerobic plates were seen with 3% sodium lactate during a storage time of 12 weeks at 4 °C [87]. Sodium lactate was also shown to inhibit the growth of several pathogens in inoculated cooked ham [88]. When samples were stored at 4 °C, 8 °C, or 10 °C, growth rates of *L. monocytogenes*, *E. coli*, or *Salmonella* decreased significantly for both 1% and 2% sodium lactate at all temperatures. However, the inhibition was more significant at lower storage temperatures [88]. The shelf life of ground beef treated with sodium lactate increased from 8 to 15 days when stored at 4 °C; aerobic plate counts, psychrotrophic counts, Enterobacteriaceae, and lactic acid bacteria counts decreased significantly compared to untreated controls [89]. Sodium lactate also effectively delayed the growth of aerobic plate counts (approximately 1.5 to 2 log reduction) in ground pork meat [90].

Barmpalia et al. [91] treated pork bologna sausage with 1.8% sodium lactate with 0.25% sodium diacetate and found that total spoilage organisms were reduced as much as 5 logs compared to the control when stored at 4 °C. Mohan and Pohlman [92] studied the effects of nine organic acids (30 g/L) and peroxyacetic acid (2 g/L) as antimicrobial treatments for beef trimmings. In general, most organic acids were effective at reducing bacterial populations, but caprylic acid produced a considerable reduction in coliforms (4.78 log reduction), *E. coli* O157:H7 (4.73 log reduction), and aerobic plate count bacteria (2.48 log reduction) [92].

The use of organic acids and their salts extends the shelf life of meat products and increases their safety due to their strong inhibitory effect against pathogenic and spoilage bacteria [93]. However, it has become apparent that some microorganisms have the means to counteract the effects of such acid-based agents, allowing them to become resistant to antimicrobial activity and to be able to survive under severe acidic conditions [94,95], leading to the emergence of acid-tolerant foodborne pathogens [96].

### 4.2. Botanicals

Botanically derived compounds, especially polyphenols, have antimicrobial effects and could be incorporated into RTE meats to reduce spoilage and increase food safety [97,98]. Polyphenols have one or more aromatic rings and one or more -OH groups, giving them their antibacterial properties [83]. These plant-based compounds can be further divided into extracts (normally hydrophilic, obtained by aqueous extractions) and essential oils (lipophilic, obtained by distillation or extraction by organic solvents) [99]. Extraction and separation of these plant substances are performed to confirm antimicrobial effects [99], but using extracts rather than plant powders also has advantages. Bioactive molecules are concentrated in the extracts so they can be used at lower concentrations than powders; the extracts are more stable, accessible, and economical to ship [83]. 

McDonnell et al. [100] evaluated the addition of vinegar, lemon, and cherry powder blend (1.5%) against the growth of *L. monocytogenes* inoculated on the surface of cured ham and deli-style turkey breast. They detected no growth of *L. monocytogenes* during 12 weeks of storage at 4 °C. Red fruit extracts (plum, red grapes, and elderberries) also inhibited pathogens, including *B. cereus*, *S. aureus*, and *E. coli*, while also increasing the growth of probiotic bacteria [101]. 

Extracts obtained from green tea, stinging nettle, and olive leaves produced a greater than 1 log CFU/g reduction in the total viable count, resulting in shelf-life extension when used in a frankfurter-type sausage [102]. Fresh chicken sausages were treated with 1% lemongrass extract, which significantly reduced the counts of total psychrotrophic and aerobic mesophilic microorganisms compared to controls [103]. Similarly, lemongrass extract extended the shelf life of cooked and shredded chicken breast and inhibited *Staphylococcus* spp., *Salmonella* spp., and coliforms [104]. 

A broad range of essential oils (EOs) and their active compounds possess antimicrobial effects. They are widely used in meat production to prevent spoilage and inhibit the growth of foodborne pathogens [105,106]. Many of these are considered GRAS [106,107], and lemon balm, basil, clove, vanilla, thyme, coriander, and others are approved for food in the U.S. [108]. Oregano oil has been proven to inhibit spoilage bacteria, extend shelf life in meat [109,110,111], and possess activity against pathogens including *Sal*. Enteritidis [112], *Sal*. Typhimurium [110,113], *St. aureus*, and *L. monocytogenes* [114]. 

Essential oils often have a strong odor and flavor, which may impact consumer acceptance and limit their use in foods [12]. In addition to their sensory effects, botanical-based antimicrobials may not mix well in food matrices or may be inactivated by standard processing techniques [115]. Other considerations include variations in the effectiveness of extracts prepared at different times from different batches of plant material and the possibility of bacteria developing resistance to the compounds [116]. There have also been concerns about the safety of these compounds since there is a potential for plants to be contaminated with heavy metals [117], mycotoxins [118], or pesticides [119]. Hurdle technology should be explored with EOs combined with other compounds or processing methods to lessen the organoleptic effects of EOs. Some suggested techniques include the encapsulation of EOs in nanostructures, which should be explored to extend the shelf life and safety of RTE meats [115]. 

### 4.3. Digestible Films and Coatings

Digestible films, also more commonly referred to as edible films, are usually cast over an inert surface, dried, and then used on food products in the form of pouches, wraps, capsules, bags, or casings [120]. Edible coatings, on the other hand, are food-grade suspensions that are sprayed or spread over the surface of a product, or the product is dipped in the suspension; when the suspension dries, it forms a transparent thin layer over the food surface and becomes a part of the final product [121]. These products can extend the shelf life of RTE meats by acting as barriers to water vapor, oxygen, and carbon dioxide, or they may also be used to carry antimicrobials [120]. These edible films and coatings can be based on lipids, proteins, or carbohydrates [120].

#### 4.3.1. Protein-Based Films and Coatings

The proteins for these products can be from both animal (casein, whey protein concentrate and isolate, collagen, gelatin, and egg albumin) and plant sources (corn, soybean, wheat, cottonseed, peanut, and rice) [120]. These protein-based films work well on hydrophilic surfaces and act as a barrier to oxygen and carbon dioxide but not water [122]. Adding hydrophobic materials such as beeswax or oils can improve the moisture barrier properties [123,124,125]. Collagen, keratin, and proteins from quinoa, milk, egg, whey, zein, and soy are widely used to make protein-based films [126]. Protein-based edible films have been investigated for salami [127], chicken breast [128,129], beef [130,131], and pork [132]. However, these films can potentially be degraded by innate enzymes in the meat or prove allergenic for some portion of the population [133].

#### 4.3.2. Lipid-Based Films and Coatings

Lipids used in edible films and coating come from both animal and plant sources. For instance, vegetable oils such as peanut, coconut, or palm oil and animal-based products including lard, butter, and fatty acids have all been utilized [134]. Fat has been used as a food coating since the 16th century [135]. Waxes and oils have been used for years as protective coatings for fresh fruits and vegetables [136]. However, wax-based fat- and oil-based coatings can be too thick, inconsistent, and greasy, or the coatings may crack [137]. There can also be a waxy or rancid taste that limits their use [137]. Therefore, lipids are usually used in a multi-component system with proteins, starch, cellulose, and their derivatives [122].

#### 4.3.3. Carbohydrate-Based Coatings and Films

Polysaccharides are carbohydrates used in coatings; they are relatively poor moisture barriers but have low permeability to O_2_ and CO_2_ and are resistant to fats and oils [138]. Cellulose, starch, pectin, seaweed extracts, gums, and chitosan can be used to make polysaccharide films [120]. These films and coatings extend the shelf life of meats by preventing dehydration, oxidative rancidity, and surface browning [120]. If these coatings are used on meat products that are smoked or steamed, the film dissolves and becomes part of the product, thus increasing yield, improving texture, and reducing loss of moisture [122]. 

Cellulose is the most abundant renewable resource on the planet, and cellulose derivatives are often used for edible films because they are biodegradable, tasteless, and odorless [139,140,141]. The most frequently utilized cellulose derivatives for edible films are methylcellulose (MC), carboxymethyl cellulose (CMC), and hydroxypropyl methylcellulose (HPMC) [142]. Films made from these cellulose derivatives are transparent and strong [143,144], and bioactive compounds can be added to impart antimicrobial properties [141]. Xie et al. [145] studied cellulose-based films with ZnO nanopillars as an antimicrobial packaging material. They concluded that these films possessed good mechanical properties, were excellent barriers to water and oxygen, and had optimal antimicrobial activity against both Gram-positive (*St. aureus*) and Gram-negative (*E. coli*) bacteria [145]. 

Starch is another abundant plant-derived polysaccharide considered cost-effective and forms desirable films [140,146,147]. Corn, potatoes, rice, cassava, tapioca, and sweet potatoes are the most common starch sources for edible films [148]. Radha Krishnan et al. [149] added clove and cinnamon essential oils to corn starch film, extending beef fillets’ shelf life. Grape juice incorporated into maize starch films increased the shelf life of chicken breast fillets [150]. Curcumin extract incorporated into a rice film increased the shelf life of chicken and fish stored at refrigeration temperatures [151]. 

Chitosan, a high-molecular-weight cationic polysaccharide obtained from the shells of crustaceans (mainly lobster and shrimp), exhibits a pronounced film-forming capacity and antimicrobial activities [141,152,153]. Chitosan-based films are transparent, flexible, tough, and very resistant to fat, oil, and oxygen but are highly sensitive to moisture [141,154]. Chitosan also possesses antimicrobial properties. Alaskan pollock sausages stored in chitosan–gelatin films had both Gram-positive and Gram-negative bacteria growth inhibited in the sausages over 42 days of storage [155].

Cyclodextrins (CDs) are water-soluble cyclic oligosaccharides made up of D-glucopyranoside units linked by α-1,4-glycosidic bonds [156]. The positions of hydroxyl groups in CDs allow the exterior to be hydrophilic while the interior is hydrophobic [157], enabling them to form inclusion complexes with hydrophobic molecules [158,159]. These CD inclusion complexes are used to maintain firmness and freshness and aid in water retention in fresh-cut fruits and vegetables [160]. Empty CDs are used to encapsulate detrimental volatile compounds inside their hollow cavity. Ethylene is the compound responsible for the ripening of fruits, but the control of ripening during shipping is desirable, and it has been successfully controlled by using CDs [161]. For example, α-CD complexes were used in one study to encapsulate ethylene to prevent the ripening of mangoes [162]. Antimicrobial compounds can also be encapsulated in CDs for slow release with various triggers. Eucalyptus essential oil was treated with CD to form an inclusion complex, and zein was added to enhance antimicrobial results; *L. monocytogenes* was reduced by 28.5% and S. aureus by 24.3% when using these films [163]. CD-based films have also been used to eliminate fish odors and preserve meat, juices, milk, beverages of all types, processed foods, ready-to-eat foods, and cheese [164].

As already noted, edible films produced from polysaccharides are suitable barriers to gas but inadequate for water vapors and have poor mechanical strength [138]. Protein-based films also have poor water vapor resistance but good mechanical strength [122]. However, lipids cannot be used to make edible films as they cannot form a cohesive structure and are used as edible coatings, primarily for fruits and vegetables, or combined with proteins or polysaccharides to make composite films [165]. Edible films have yet to be commercialized for various reasons, not the least of which is that edible films are not as strong as plastic films and do not elongate well [166]. Jeevahan et al. [166] have written an excellent article addressing the problems with the commercialization of edible films, addressing six problem areas: (a) functional properties, (b) film making and drying methods, (c) nanotechnology on edible films, (d) lack of knowledge, and (e) consumer acceptance.

### 4.4. Bacteriocins: Sources and Applications for RTE Meats in the U.S.

Bacteriocins are peptides produced by certain bacteria that can exhibit activity against Gram-positive and Gram-negative bacteria [167,168]. They are known to be safe for human consumption because they are inactivated by proteases in the gastrointestinal tract (GIT) without affecting the normal microbiota of the GIT [169]. An attractive feature is that target bacteria do not form resistance against them [170]. There are three different approaches to the use of bacteriocins in food: (1) direct addition of the bacteriocin into the product; (2) incorporation of the bacteriocin into an edible film; (3) inclusion of bacteriocin-producing cultures into the formula or in a coating [169].

*Lactococcus lactis* subsp. *lactis* produces a bacteriocin, nisin, which has been found to inhibit the growth of *Clostridium* and *Bacillus* [171]. Many countries, including the U.S., allow the use of nisin in food products such as milk, processed cheese, grated cheese, dairy products, canned vegetables, soups, and meat, as well as brewery products and mayonnaise [172,173,174]. However, the use of nisin in meat is limited because it has low solubility in meat products, innate enzymes in the meat may destroy nisin, and it has shown limited effectiveness against several meat spoilage organisms [175].

Pediocin PA-1/AcH, produced by *Pediococcus acidilactici*, is another bacteriocin that can be used in dried sausages and fermented meat products [168,170,175]. *P. acidilactici* MCH14, which produces pediocin PA-1, was incorporated into dry fermented sausages and was shown to inhibit the growth of both *L. monocytogenes* and *Cl. perfringens* [176]. Strain *P. pentosaceus* BCC3772, which produces pediocin PA-1/AcH, inhibited *L. monocytogenes* during the fermentation of a Thai pork sausage without significantly altering its odor or taste [177]. Pediocin and nisin were used in vacuum-packed sliced ham and reduced the counts of *Lactobacillus sakei*, an important spoilage organism [178]. Pediocin PA-1 minimizes the growth of spoilage microorganisms during the storage of meat products; it is active at low pH and acts synergistically with lactate or organic acids [179]. 

A bacteriocin produced by *Leuconostoc mesenteroides* ssp. *mesenteroides* IMAU: 10231 was incorporated into a fermented sausage product, reducing the growth of *L. monocytogenes* during 21 days of refrigerated storage [180]. This strain of *Leuconostoc* produces CO_2_ during fermentation, which means the bacteriocin itself must be used rather than the bacteria being incorporated as a part of the culture [181]. However, a homofermentative lactic acid bacteria (LAB) that produces lactic acid instead of CO_2_ could be used as a starter culture [169]. For example, *Weissella paramesenteroides* DX produces the bacteriocin weissellin A. When this organism is grown in oxygen, sodium nitrite inhibits the bacteriocin production [181]. However, in anaerobic conditions such as those found when fermenting meat sausages, sodium nitrite does not inhibit the production of bacteriocin, making this organism an optimal candidate as a starter culture [181]. The bacteriocin BacFL31 produced by *Enterococcus faecium* sp. FL 31 inhibited the growth of spoilage microorganisms, *L. monocytogenes*, and *Sal*. Typhimurium in ground turkey meat during refrigerated storage [182]. 

Direct application of bacteriocins in the meat matrix reduces the antimicrobial activity of some other bacteriocins [183]. To avoid reductions in the efficacy of other bacteriocins, an alternative application method is to incorporate the bacteriocin into the food packaging material to serve as an active packaging component, therefore reducing the direct contact of the bacteriocin with the food product [169]. This approach allows the use of a smaller quantity of bacteriocins and controlled release, as the activity is on the product’s surface [184]. Over the past ten years, nisin has been widely utilized as an active packaging component [185]. Nisin used in various films and coatings has been shown to reduce the growth of *L. monocytogenes*, *B. thermosphacta*, *Enterobacteriaceae*, and spoilage LAB in raw meat, sliced ham, and ground beef [130,186,187,188]. Pediocin incorporated into cellulose-based packages also reduced the growth of *L. monocytogenes* in sliced ham, turkey, and beef [189,190]. Marcos et al. [191] demonstrated the increased safety of sliced ham by delaying or reducing the growth of *L. monocytogenes* using enterocins incorporated into alginate, zein, or polyvinyl alcohol-based biodegradable film.

Despite the success of nisin and pediocin, the use of bacteriocins in foods may be limited by their poor solubility and the uneven distribution and partitioning in the food matrix [169,192]. Their narrow spectrum of activity means that specific spoilage organisms must be identified and bacteriocins discovered that inactivate these bacteria [193]. Purification of bacteriocins can be difficult and expensive [193].

### 4.5. Bacteriophage for RTE Meats in the U.S.

Bacteriophages (phages), viruses that infect bacteria, are ubiquitous in the world, existing everywhere that there are bacteria [194,195,196]. Bacteriophages have been studied for use as antimicrobials to increase food safety [197]. Since bacteriophages are specific for a particular bacterium, most studies in RTE meats and bacteriophages have been conducted on pathogens, especially *L. monocytogenes* [198]. The U.S. Food and Drug Administration (FDA) has approved two bacteriophage preparations, using phages P100 and LMP-102, as food ingredients to control *L. monocytogenes* [199]. 

Alves et al. [200] incorporated bacteriophage IBB-PF7A in a sodium alginate film to limit meat spoilage caused by *Pseudomonas fluorescens*. The number of *Ps. fluorescens* organisms decreased by 2 Logs CFU/cm^2^ during the first two days of refrigerated storage and then only dropped by 1 Log CFU/cm^2^ over the next five days [200]. This highlights one of the problems with using bacteriophage in food products: their stability [200]. 

A more significant problem is the specificity of phages [198]. To target specific spoilage organisms, there is a need to definitively identify most, if not all, members of the microbial community of meat and meat products, especially regarding spoilage. Another limitation is that phages cannot be combined with other methods because the other techniques would inactivate the phages [201]. There are two types of bacteriophages, lytic and lysogenic, which also challenge their use [195]. Lytic phages take over the protein synthesis of the target cell and produce more phages, which then burst the target cell to infect other bacteria [195]. Lysogenic phages, on the other hand, integrate themselves into the host cell’s chromosome, making lytic phages the most suitable for food applications [195,201]. Bacteriophages cannot diffuse throughout food and are limited to surface application [201]. 

## 5. Conclusions and Future Prospects

Table 3 compares the advantages and disadvantages of these natural antimicrobials; using any of these natural antimicrobials as clean-label alternatives can effectively increase the shelf life and safety of foods. In many cases, however, dosages must be adjusted to provide a practical antimicrobial effect without affecting the sensory qualities of the food. Many of these natural antimicrobials can be chemically degraded during processing at high temperatures or high-pressure treatments. It is also important to note that the effectiveness of these antimicrobials is always higher in the lab than in an actual food matrix [202], making it essential that these compounds be tested in actual foods [203,204]. More research is needed to assess the synergistic effect of different natural antimicrobials, their optimum concentrations, and their toxicity. Molecular studies of the microbiome of these meat products and the processing environment could help in these evaluations [205].

The clean-label trend is here to stay as consumers exhibit more interest in what they perceive as more healthy foods. However, a key hazard of this movement is the removal of ingredients that have been a foundation of food safety in RTE meats, such as lactate and diacetate. Consumer-friendly ingredients must continue to foster food safety for the consumer and adhere to regulatory guidelines. Past research has primarily focused on pathogens, but more research is needed about meat spoilage and product shelf life. In addition, a better understanding of the mechanism(s) associated with these clean-label antimicrobial compounds is essential. Once antimicrobial mechanism(s) can be identified, opportunities for combinations of mechanistically different compounds that target a broader spectrum of microorganisms could lead to optimal multiple hurdle applications in RTE meats. 

Various natural antimicrobials are currently being used and are potentially available for application in RTE meats; each has specific attributes that support their utility for practical applications (Table 3). Using any of these natural antimicrobials as clean-label alternatives can effectively increase food’s shelf life and safety. In many cases, however, dosages must be adjusted to provide a practical antimicrobial effect without affecting the sensory qualities of the food. Many of these natural antimicrobials can be chemically degraded during processing at high temperatures or high-pressure treatments. It is also important to note that the effectiveness of these antimicrobials is always higher under laboratory conditions than in an actual food matrix [202], making it essential that these compounds be tested in actual foods [203,204]. More research is needed to assess the synergistic effect of different natural antimicrobials, their optimum concentrations, and their toxicity. Molecular studies based on 16S rDNA microbiome sequencing of these meat products and the processing environment could help in these evaluations by assessing overall microbial community responses to these antimicrobials [205]. This microbial analysis would help determine which natural antimicrobials possess the most broad-spectrum efficacy in RTE meats.

One of the major concerns expressed by food processors about the “clean-label” movement is the needless elimination of ingredients that have played essential roles in terms of food preservation, raising concerns about consumer health, food safety, and shelf life. Food additives that are considered acceptable by consumers must also continue to protect those consumers while still obeying regulatory guidelines. More research is needed to elucidate the factors that cause consumers to see an additive as “natural” as compared to “synthetic”. It is imperative for the meat industry to bolster consumer trust by incorporating transparency into the food chain.

## Figures and Tables

**Table 1 foods-13-00464-t001:** Certain nonorganic, nonagricultural antimicrobials, additives, and preservatives designated by the USDA as allowable in organic RTE meat products ^A^.

Substance	Synthetic	Nonsynthetic	Use	Specifications and Restrictions
Acids, citric and lactic		+	Acidifier, preservative, flavoring, antimicrobial [23]	When produced through microbial fermentation.
Food-grade microorganisms		+	Fermentation and sensory development, antimicrobial	Bacteria, fungi.
Yeast		+	Fermentation and sensory development	Must be organic if end use is for human consumption, nonorganic if not commercially available. Cannot be grown on a petrochemical substrate or sulfite waste liquor. If smoked yeast, nonsynthetic smoke flavoring process must be documented.
Acidified sodium chloride	+		Antimicrobial	Secondary direct antimicrobial food treatment and indirect food contact surface sanitizing. Acidified with citric acid only.
Ascorbic acid	+		Antioxidant, antimicrobial [24]	
Calcium phosphates	+		Fluid retention [25]	Monobasic, dibasic, and tribasic.
Calcium hypochlorite	+		Antimicrobial [26]	
Collagen gel	+		Casing material	Casing, may be used when organic collagen gel is not commercially available.
Ozone	+		Antimicrobial	
Peracetic acid/peroxyacetic acid	+		Antimicrobial	For use in wash and/or rinse water according to FDA limitations; for use as a sanitizer on food contact surfaces.
Phosphoric acid	+		Antimicrobial (surfaces only)	Cleaning of food contact surfaces and equipment only.
Potassium citrate	+		Fluid retention [27]	
Potassium lactate	+		Fluid retention [27]	Antimicrobial agent and pH regulator only.
Potassium phosphate	+		Fluid retention	For use only in products labeled “made with organic…”
Sodium citrate	+		Acidifier, preservative, flavoring, antimicrobial	
Sodium lactate	+		Acidifier, preservative, flavoring, antimicrobial	Antimicrobial agent and pH regulator only.
Tocopherols	+		Antioxidant	Derived from vegetable oil when rosemary extracts are not a suitable alternative.
Xanthan gum	+		Thickening agent, emulsion stabilizer, fat replacer [28]	

^A^ For more information on these products, please see 7 CFR § 205.605.

**Table 2 foods-13-00464-t002:** Nitrate and nitrite contents of edible components of vegetables.

Vegetable Types and Varieties	Nitrite	Nitrate
	mg/100 g fresh weight	mg/100 g fresh weight
Root vegetables		
Carrot	0.002–0.023	92–195
Mustard leaf	0.012–0.064	70–95
Green vegetables		
Lettuce	0.008–0.215	12.3–267.8
Spinach	0–0.073	23.9–387.2
Cabbage		
Chinese cabbage	0–0.065	42.9–161.0
Bok choy	0.009–0.242	102.3–309.8
Cabbage	0–0.041	25.9–125.0
Cole	0.364–0.535	76.6–136.5
Melon		
Wax gourd	0.001–0.006	35.8–68.0
Cucumber	0–0.011	1.2–14.3
Nightshade		
Eggplant	0.007–0.049	25.0–42.4

Reprinted from [33] with permission from Elsevier.

**Table 3 foods-13-00464-t003:** Clean-label interventions: advantages and disadvantages.

Intervention	Advantages	Disadvantages
Organic acids	GRAS status Low cost Easy to use	Some microorganisms can develop acid-tolerant strains
Plant-based compounds	Broad range of antimicrobial effects	Unstable Limited dispersion in food matrices Negative sensory impact on food
Edible films and coatings	Reduce escape of moisture and volatiles Inhibit biochemical breakdown Improve food appearance Improve nutritional value	May be degraded by native enzymes in food May be allergenic
Bacteriocins	Inactivated by enzymes in human stomach	May have low solubility in food matrix May be degraded by innate enzymes in the food Narrow spectrum of activity Purification is difficult and expensive
Bacteriophages	Ease of use against pathogens	Specificity: must know target organism Cannot be combined with other methods Limited to surface application

## Data Availability

The data presented in this study are available on request from the corresponding author. The data are not publicly available due to privacy restrictions.

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
