# Peer review of "Alternative Additives for Organic and Natural Ready-to-Eat Meats to Control Spoilage and Maintain Shelf Life: Current Perspectives in the United States"

_foods, 2024, doi:10.3390/foods13030464_

Round 1

Reviewer 1 Report

Comments and Suggestions for Authors

The review is well-crafted with no significant issues. However, the following points need to be solved:

  1. 1. In recent applications, there may have been instances of utilizing cyclodextrin-based inclusion complexes within the realm of carbohydrate-based coatings or films. The author is requested to incorporate this aspect into the relevant chapter.

  2. 2. To augment the impact of the review, it is recommended to include a section on future scope or prospects.

  3. 3. I would like to point out that, in my opinion, some references may not be acceptable, such as those labeled 2, 5, 9, etc. Please review and substitute them accordingly.

Author Response

  1. In recent applications, there may have been instances of utilizing cyclodextrin-based inclusion complexes within the realm of carbohydrate-based coatings or films. The author is requested to incorporate this aspect into the relevant chapter.

Added. Please see L 413-428.

  1. To augment the impact of the review, it is recommended to include a section on future scope or prospects.

This has been added

  1. I would like to point out that, in my opinion, some references may not be acceptable, such as those labeled 2, 5, 9, etc. Please review and substitute them accordingly.

Author response: We feel these references are relevant and should be kept as they reflect consumer perceptions and anecdotes that set up the rationale for retail market movement towards clean labels.

Reviewer 2 Report

Comments and Suggestions for Authors

The subject of the article itself its quite interesting. However there is a quite similar article already published in Foods on the same subject see:  Santiesteban-López NA, Gómez-Salazar JA, Santos EM, Campagnol PCB, Teixeira A, Lorenzo JM, Sosa-Morales ME, Domínguez R. Natural Antimicrobials: A Clean Label Strategy to Improve the Shelf Life and Safety of Reformulated Meat Products. Foods. 2022 Aug 29;11(17):2613. doi: 10.3390/foods11172613. PMID: 36076798; PMCID: PMC9455744.

Author Response

The subject of the article itself its quite interesting. However there is a quite similar article already published in Foods on the same subject see:  Santiesteban-López NA, Gómez-Salazar JA, Santos EM, Campagnol PCB, Teixeira A, Lorenzo JM, Sosa-Morales ME, Domínguez R. Natural Antimicrobials: A Clean Label Strategy to Improve the Shelf Life and Safety of Reformulated Meat Products. Foods. 2022 Aug 29;11(17):2613. doi: 10.3390/foods11172613. PMID: 36076798; PMCID: PMC9455744.

Author response: We disagree. While some of the food additives that are discussed are similar our focus is on Ready to Eat Meats from a United States perspective. To reflect this distinction, we have changed the title of our review to “Alternative Additives for Organic and Natural Ready-to-Eat Meats to Control Spoilage and Maintain Shelf Life: Current Perspectives in the United States”

Also we made similar changes in some of the subheadins to reflect the U.S. focus of our review and the U.S. regulations regarding these food additives for RTE meats. Likewise the microbial spoilage ecology is focused on RTE spoilage. The Santiesteban-López article also concentrates on antimicrobials, which is not the case with our review.

Reviewer 3 Report

Comments and Suggestions for Authors

The review article provides a good overview of the challenges and opportunities in developing natural antimicrobials for ready-to-eat (RTE) meat products. The demand for 'clean label' foods has increased, leading to the need for natural antimicrobials in ready-to-eat (RTE) meats. Traditional additives are being replaced by compounds like plant-based antimicrobials, bacteriocins, and bacteriophages. These antimicrobials can reduce spoilage organisms and prevent the growth of foodborne pathogens. However, challenges include their limited dispersion in food matrices, negative sensory impact, degradation by native enzymes, narrow spectrum of activity, and difficulties in purification. Further research is needed to understand their mechanisms and explore combinations for optimal applications in RTE meats.

Author Response

The review article provides a good overview of the challenges and opportunities in developing natural antimicrobials for ready-to-eat (RTE) meat products. The demand for 'clean label' foods has increased, leading to the need for natural antimicrobials in ready-to-eat (RTE) meats. Traditional additives are being replaced by compounds like plant-based antimicrobials, bacteriocins, and bacteriophages. These antimicrobials can reduce spoilage organisms and prevent the growth of foodborne pathogens. However, challenges include their limited dispersion in food matrices, negative sensory impact, degradation by native enzymes, narrow spectrum of activity, and difficulties in purification. Further research is needed to understand their mechanisms and explore combinations for optimal applications in RTE meats.

Authors thank you for your time

Reviewer 4 Report

Comments and Suggestions for Authors

Overall, the article is well-written and covers an interesting and important topic. It provides valuable insights into the evolving landscape of food additives, particularly in the context of clean label trends. The discussion on various clean label additives and their implications for the quality and safety of RTE meats is thorough and informative. Further comments and suggestions for improvements are detailed below:

  1. Lines 34 – 35: The article should specify the primary additives used in meat products.
  2. Lines 124 – 155: A deeper discussion on the use of natural nitrates (from vegetables) as opposed to synthetic nitrates/nitrites, focusing on the impact of natural nitrates on nitrosamine formation, is recommended. Relevant studies using natural extracts as nitrate sources in RTE products should be cited.
  3. Line 167: A sentence needs correction.
  4. Lines 165 – 185: The section should also address dry sausages.
  5. Line 193: It should be clarified that cooking does not eliminate all bacteria.
  6. Line 218: Correct the use of 'can, cans...'
  7. Line 229: Mention various other products as well.
  8. Line 281: Discuss whether sodium lactate can be considered a clean label additive.
  9. Lines 301 – 306: Explain the antimicrobial mechanism of organic acids and their salts.
  10. Lines 523 – 551: Synthesize the conclusion without repeating previous information.

Additionally, it is suggested that a general figure illustrating all clean label additives be included to enhance the visual appeal and comprehension of the article.

Author Response

Overall, the article is well-written and covers an interesting and important topic. It provides valuable insights into the evolving landscape of food additives, particularly in the context of clean label trends. The discussion on various clean label additives and their implications for the quality and safety of RTE meats is thorough and informative. Further comments and suggestions for improvements are detailed below:

    Lines 34 – 35: The article should specify the primary additives used in meat products.

We emphasized the additives in this review that were either commercialized (bacteriocins for example) or extensively researched.  We are not sure what the criteria would be for “primary”.

    Lines 124 – 155: A deeper discussion on the use of natural nitrates (from vegetables) as opposed to synthetic nitrates/nitrites, focusing on the impact of natural nitrates on nitrosamine formation, is recommended. Relevant studies using natural extracts as nitrate sources in RTE products should be cited.

The subject of natural nitrates/nitrites deserves a review all its own. We have touched on the subject here with the conclusion that consumers regard any added nitrates/nitrites with suspicion.

    Line 167: A sentence needs correction. Sentence fragment deleted

    Lines 165 – 185: The section should also address dry sausages.

Dry sausages are discussed in in this section

    Line 193: It should be clarified that cooking does not eliminate all bacteria.

Clarified

    Line 218: Correct the use of 'can, cans...' Corrected

    Line 229: Mention various other products as well. Uncertain what it being asked for

    Line 281: Discuss whether sodium lactate can be considered a clean label additive.

This is addressed in the paper  

    Lines 301 – 306: Explain the antimicrobial mechanism of organic acids and their

salts. Added

    Lines 523 – 551: Synthesize the conclusion without repeating previous information.

Rewritten

Additionally, it is suggested that a general figure illustrating all clean label additives be included to enhance the visual appeal and comprehension of the article.

It would be difficult to construct an all inclusive figure of all clean label additives

Round 2

Reviewer 3 Report

Comments and Suggestions for Authors

The manuscript is suitable for publication in the present form